# ReBotNet: Fast Real-time Video Enhancement

## Abstract

Most video restoration networks are slow, have high computational load, and can't be used for real-time video enhancement. In this work, we design an efficient and fast framework to perform real-time video enhancement for practical use-cases like live video calls and video streams. Our proposed method, called **Re**current **Bot**tleneck Mixer **Net**work (**ReBotNet**), employs a dual-branch framework. The first branch learns spatio-temporal features by tokenizing the input frames along the spatial and temporal dimensions using a ConvNext-based encoder and processing these abstract tokens using a bottleneck mixer. To further improve temporal consistency, the second branch employs a mixer directly on tokens extracted from individual frames. A common decoder then merges the features form the two branches to predict the enhanced frame. In addition, we use a recurrent training approach where the last frame's prediction is leveraged to efficiently enhance the current frame while improving temporal consistency. To evaluate our method, we curate two new datasets that emulate real-world video call and streaming scenarios, and show extensive results on multiple datasets where ReBotNet outperforms existing approaches with lower computations, reduced memory requirements, and faster inference time.

## 1 Introduction

Video enhancement has several use-cases in surveillance Shen et al. (2022); Ding et al. (2020); Rajan & Binu (2016), cinematography Wan et al. (2022); Iizuka & Simo-Serra (2019), medical imaging Katsaros et al. (2022); Stetson et al. (1997), virtual reality Wang & Zhao (2022); Pearl et al. (2022); Han et al. (2007); Vassallo et al. (2018), sports streaming Chang et al. (2001); Zhao (2022), and video streaming Zhang et al. (2020). It also facilitates downstream tasks such as analysis and interpretation Rao & Chen (2012), e.g., it improves accuracy of facial recognition algorithms, allows doctors to diagnose medical conditions more accurately, and helps in better sports analysis by understanding player movements and tactics. Also, the recent rise of hybrid work has led to an immense increase in video conferencing, where poor video quality due to a low quality camera, poor lighting conditions, or a bad network connection can obscure non-verbal cues and hinder communication and increase fatigue Döring et al. (2022). Thus, there lies a significant interest in developing methods that can perform real-time video enhancement.

Unlike individual restoration tasks like denoising Elad et al. (2023); Tian et al. (2019), deblurring Zhang et al. (2022b); Sahu et al. (2019), super-resolution Wang et al. (2020b); Liu et al. (2022a) which focus on restoring videos affected by a single degradation; generic video enhancement techniques focus on improving the overall quality of videos and make them look better Xue et al. (2019). In this setup, there are multiple degradations that can interact in a complex way, e.g., compression of a noisy video, camera noise, motion blur etc. mirroring the real world scenarios. Video restoration methods can be adopted for video enhancement by training on a dataset that includes multiple degradations. However, from our experiments we found that they are computationally complex and have a high inference time and are not suitable for real-time applications. Also, many methods take past and future frames as input which will introduce latency in streaming video.

In this paper, we develop an efficient video enhancement network that achieves state of the art results and enables real-time processing. At the core of our method is a novel architecture using convolutional blocks at early layers and MLP-based blocks at the bottleneck. Following Srinivas et al. (2021) which uses a convolutional encoder for initial feature extraction followed by a transformer network in the bottleneck, we propose a network where the initial layers extract features using ConvNext Liu

et al. (2022b) blocks and a bottleneck consisting of MLP mixing blocks Tolstikhin et al. (2021). This design avoids quadratic computational complexity of vanilla attention Vaswani et al. (2017), while maintaining a good performance. We also tokenize the input frames in two different ways to enable the network to learn both spatial and temporal features. Both these token sets are passed through separate mixer layers to learn dependencies between these tokens. We then use a simple decoder to predict the enhanced frame. To further improve efficiency and improve temporal consistency, we exploit the fact that real world videos typically have temporal redundancy implying that the prediction from previous frame can help current frame's prediction. To leverage this redundancy, we use a frame-recurrent training setup where the previous prediction is used as an additional input to the network. This helps us carry forward information to the future frames while being more efficient than methods that take a stack of multiple frames as input. We train our proposed network in this recurrent way and term our overall method **Re**current **Bot**tleneck Mixer **Net**work (ReBotNet).

To evaluate our method, we curate and introduce two new datasets for video enhancement. The existing video restoration datasets focus on a single task at a time, e.g., denoising (DAVIS Khoreva et al. (2019), Set8 Tassano et al. (2019), etc.), deblurring (DVD Su et al. (2017), GoPro Nah et al. (2017), etc.), and super-resolution (REDS Nah et al. (2019), Vid4 Liu & Sun (2013), Vimeo-90k-T Xue et al. (2019), etc.). These datasets do not emulate the real-world case where the video is degraded by a mixture of many artifacts. Also, rise in popularity of video conferencing calls for datasets that have semantic content similar to a typical video call. Single image enhancement methods are often studied on face images Liu et al. (2018); Karras et al. (2019) because human perception is very sensitive to even slight changes in faces. However, a comparable dataset for video enhancement research has yet to be established. To this end, we curate a new dataset called *PortraitVideo* that contains cropped talking heads of people and their corresponding degraded version obtained by applying multiple synthetic degradations. The second dataset, called *FullVideo*, contains a set of degraded videos without face alignment and cropping. We conduct extensive experiments on these datasets and show that we obtain better performance with less compute and faster inference than recent video restoration frameworks. In particular, our method is 2.5x faster while either matching or in some cases obtaining a PSNR improvement of 0.2 dB over previous SOTA method. This shows the effectiveness of our proposed approach and opens up exciting possibilities of deploying them in real-time applications like video conferencing.

In summary, we make the following major contributions:

- We work towards **real-time** video enhancement, with a specific focus on practical applications like video calls and live streaming.
- We propose a new method: Recurrent Bottleneck Mixer Network (ReBotNet) , an efficient deep neural network architecture for real-time video enhancement.
- We curate two new video enhancement datasets: *PortraitVideo*, *FullVideo* which emulate practical video enhancement scenarios.
- We perform extensive experiments where we find that ReBotNet matches or exceeds the performance of baseline methods while being significantly faster.

## 2 RELATED WORKS

Image and video restoration Dong et al. (2014); Fan et al. (2017; 2020); Zhou et al. (2019); Yasarla et al. (2020); Yi et al. (2021; 2019) is a widely studied topic where CNN-based methods have been dominating over the past few years. For video restoration, most CNN-based methods take a sliding window approach where a sequence of frames are taken as input and the center frame is predicted Su et al. (2017); Tassano et al. (2019). To address motion between frames, many methods explicitly focus on temporal alignment Chan et al. (2021a); Zhu et al. (2022); Chan et al. (2021b), with optical flow being a popular alignment method. Dynamic upsampling filters Jo et al. (2018), spatio-temporal transformer networks Kim et al. (2018), and deformable convolution Tian et al. (2020) have been proposed for multi-frame optical flow estimation and warping. Aside from sliding window approaches, another widely used technique is a recurrent framework where bidirectional convolutional neural networks warp the previous frame prediction onto the current frame Chan et al. (2021a; 2022a); Fuoli et al. (2019); Haris et al. (2019); Huang et al. (2015). These recurrent methods usually use optical flows to warp the nearby frames to create the recurrent mechanism. Unlike these works that require optical flow, we develop a simple and efficient frame-recurrent setup with low

computational overhead. As most of these methods use synthetic datasets, recent works have looked into adopting these methods for real-world application Yang et al. (2021); Chan et al. (2022b) One recent work attempted to solve multiple degradation problem that includes blur, aliasing and low resolution with one model Cao et al. (2022) but it is still computationally intensive.

Video restoration transformer (VRT) introduced a parallel frame prediction model leveraging long-range temporal dependency modelling abilities of transformers Liang et al. (2022a). Recurrent video restoration transformer (RVRT) Liang et al. (2022b) introduced a globally recurrent framework with processing neighboring frames. At the time of writing, it is worth mentioning that RVRT stands as the SOTA method for most video restoration datasets. Unlike above methods, we focus on developing real-time solutions for generic video enhancement with a focus on practical applications like live video calls.

## 3 METHOD

### 3.1 RECURRENT BOTTLENECK MIXER

Transformers Dosovitskiy et al. (2020) form the backbone of current state of the art video restoration methods Liang et al. (2022a;b) due to their ability to model long-range dependencies but suffer from high computational cost due to the quadratic complexity of attention mechanism. Attention with linear complexity Wang et al. (2020a); Zhang et al. (2022a); Katharopoulos et al. (2020) reduces performance while still not achieving real-time inference. On the other hand, Zhai et al. (2021); Liu et al. (2021); Wang et al. (2022) show that attention can be replaced by other mechanisms with marginal regression in quality, e.g., Tolstikhin et al. (2021) replaces self-attention with much more efficient token mixing multi-layer perceptrons (MLP-Mixers). Mixers have been subsequently shown to be useful for multiple tasks Touvron et al. (2022); Valanarasu & Patel (2022); Yu et al. (2022); Qiu et al. (2022); Tu et al. (2022); Ma et al. (2022). However, Mixers do not work out-of-the box for video enhancement, as (i) they lead to a significant regression in quality (in our experiments in supplementary material) compared to transformer-based approaches, and (ii) while more efficient, they still do not yield real-time inference on high resolution imagery. Also, videos are processed using transformers by either representing them as tubelets or patch tokens Arnab et al. (2021). However, tubelet tokens Arnab et al. (2021) and image tokens Dosovitskiy et al. (2020) can be complementary with different advantages and disadvantages. Tubelet tokens can compactly represent spatio-temporal patterns. On the other hand, image tokens or patch tokens extracted from an individual frame represents only spatial features without spending capacity on modeling motion cues. These issues motivate us in developing a new backbone for video enhancement with mixers at its core while combining tubelets and image tokens in a single efficient architecture.

An overview of ReBotNet can be found in Fig. 1. ReBotNet takes two inputs: the previous predicted frame ($y_{t-1}$) and the current frame ($x_t$). We use an encoder-decoder architecture where the encoder has two branches. The first branch focuses on spatio-temporal mixing where we tokenize the input frames as tubelets and then process these spatio-temporal features using mixers in the bottleneck. The output features of this mixer block has information processed along both the spatial and temporal dimensions. The second branch extracts just the spatial features using linear layers from individual frames. These tokens contain only spatial information as the frames are processed independently. These spatial features are forwarded to another mixer bottleneck block which learns the inter-dependencies between these tokens. This mixer block captures temporal information by extracting the relationship between tokens from individual frames, thereby encoding the temporal dynamics. The resultant features from both branches are added and are forwarded to a decoder which consists of transposed convolutional layers to upsample the feature maps to the same size as of the input. We output a single prediction image ($y_t$) which is the enhanced image of the current frame ($x_t$).

### 3.2 ENCODER AND TOKENIZATION

Tokenization is an important step in pre-processing data for transformer-based methods as it allows the model to work with the input data in a format that it can understand and process Qian et al. (2022). For our network, we use two different ways of doing tokenization: i) tubelet tokens and ii) image tokens.

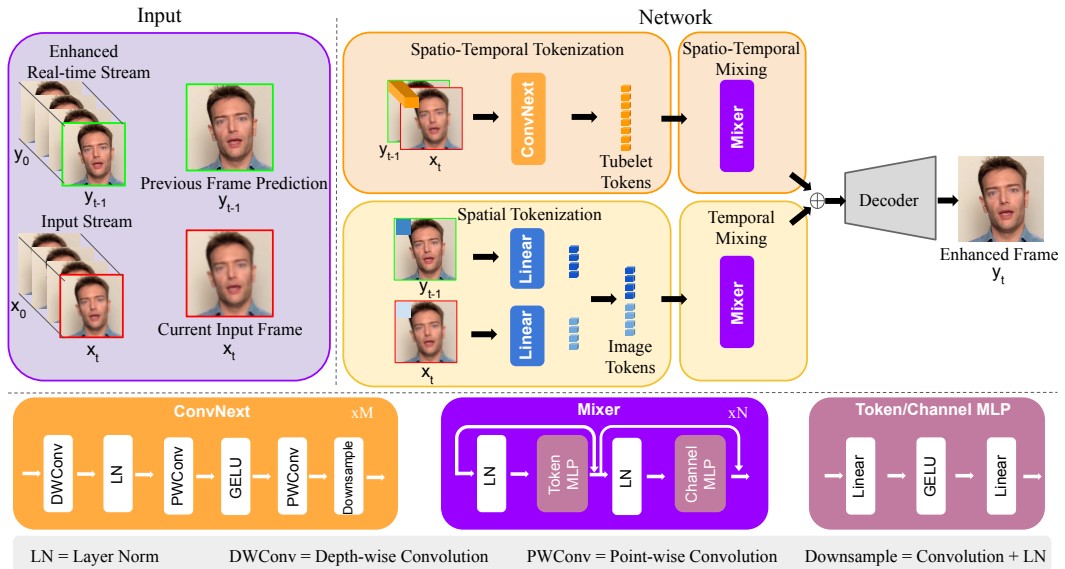

Figure 1: Overview of the proposed Recurrent Bottleneck Mixer Network. The inputs to the network are the previous frame prediction and the current input frame. These are tokenized in two different ways: Tubelet tokens and image tokens . The tubelet tokens are processed using a Mixer to learn spatio-temporal features while image tokens are processed using a Mixer to learn temporal features. These features are passed through an upsampling decoder to get the output enhanced frame.

**Branch 1 - Tubelet tokens:** Tubelet tokens are extracted across multiple frames, in our case, the current frame and the previous predicted frame, and encode spatio-temporal data. Convolutional layers can be advantageous in extracting tokens as they can capture more informative features compared to linear layers due to their inductive bias Xie et al. (2021). Hence, we stack the input images: $y_{t-1}$, $x_t$ across the channel dimension and directly forward them to ConvNext blocks Liu et al. (2022b), which are more efficient and powerful than vanilla convolutional layers. Each ConvNext block consists of a depth-wise convolution layer Chollet (2017) with kernel size of $7 \times 7$, stride 1 and padding 3 followed by a layer normalization Ba et al. (2016) and a point-wise convolution function. The output of this is activated using GeLU Hendrycks & Gimpel (2016) activation and then forwarded to another point-wise convolution to get the output. More details of this why this exact setup is followed can be found in supplementary. We also have downsampling blocks after each level in the ConvNext encoder. These tubelet tokens compromise the first branch of ReBotNet where we do spatio-temporal mixing. These tokens are further processed using a bottleneck mixer to enhance the features and encode more spatio-temporal information.

**Branch 2 - Image tokens:** The individual frames $y_{t-1}$, $x_t$ are from different time steps. Although tubelet tokens encode temporal information, learning additional temporal features can only improve the stability of the enhanced video and help get clearer details for enhancement. We do this by extracting individual image tokens and learn the correspondence between them. To this end, we tokenize the images individually by converting them into patches and using linear layers like in ViT Dosovitskiy et al. (2020). In this branch, we use linear layers instead of ConvNext blocks for the sake of efficiency although ConvNext blocks extract more representative and useful features. The main goal of this block is to ensure that the temporal features of the input data remain consistent. Note that high quality spatial features necessary for enhancing spatial quality, is handled in the first branch. To this end, the mixer bottleneck learns to encode the temporal information between these image tokens extracted from individual frames.

We ensure that the tubelet tokens and image tokens have the same dimensions of $N \times C$, where $N$ is the number of tokens and $C$ is the number of channel embeddings. To achieve this, we max-pool image tokens to match the dimensions of tubelet tokens.

### 3.3 BOTTLENECK

The bottleneck of both the branches consist of mixer networks with the same basic design. The mixer network takes in tokens $T$ as input and processes them using two different multi-layer perceptrons (MLPs). First, the input tokens are normalized and then mixed across the token dimension. The process can be summarized as:

$$T_{TM} = MLP_{TM}(LN(T_{in})) + T_{in}, \tag{1}$$

where $T_{TM}$ represents the tokens extracted after Token Mixing (TM), $T_{in}$ represents the input tokens, and LN represents layer normalization Ba et al. (2016). Note that there is also a skip connection between the input to the mixer and the output from token mixing MLP. Token mixing encodes the relationship between individual tokens. Afterwards, the tokens are flipped along the $C$ axis and fed into another MLP to learn dependencies in the $C$ dimension Tolstikhin et al. (2021). This is called channel mixing and is formulated as follows:

$$T_{out} = MLP_{CM}(LN(T_{TM})) + T_{TM}, \tag{2}$$

where $T_{out}$ represents the output tokens and $CM$ denotes channel mixing. The MLP block comprises of two linear layers that are activated by GeLU Hendrycks & Gimpel (2016). The initial linear layer converts the number of tokens/channels into an embedding dimension, while the second linear layer brings them back to their original dimension. The selection of the embedding dimension and the number of mixer blocks for the bottleneck is done through hyperparameter tuning.

### 3.4 RECURRENT TRAINING

Recurrent setups generally refer to a type of configuration or arrangement that is repeated or ongoing Sherstinsky (2020). In real-time video enhancement, the original video stream has to be enhanced on-the-fly which means we have the information of all the enhanced frame till the current time instance. The enhanced frames from the previous time step has valuable information that could be leveraged for the current prediction for increased efficiency. Leveraging previous frame prediction can also help in increasing the temporal stability of the predictions as the current predictions gets conditioned on the previous predictions. Although it is possible to use multiple previous frames in a recurrent setup, we have chosen to only use the most recent prediction for the sake of efficiency.

In the following, we elucidate how we leverage the recurrent setup to output the enhanced frame. Let us define the original input stream as $X = \{x_0, x_1, ...., x_t\}$ where $X$ denotes the video and $x$ denotes the individual frames. The frames start from the initial frame $x_0$ to the current time frame $x_t$. Similarly, we also define the enhanced video stream represented as $Y = \{y_0, y_1, ...., y_{t-1}\}$ where $Y$ denotes the enhanced video stream and $y$ denotes the individual enhanced frames. These enhanced frames go from the initial time step $y_0$ to the previous time frame $y_{t-1}$. So, to find the enhanced prediction of the current frame $y_t$, we make use of current degraded frame $x_t$ and the previous enhanced frame $y_{t-1}$. These images are sent to the network to output $y_t$. In the context of training, a single feed forward step involves using the input values $x_t$ and $y_{t-1}$ to make a prediction for the output value $y_t$. When processing a video, multiple feed forward steps are used in a sequential manner to predict the output values for all frames in the video. Similarly, during backpropagation, the gradients are propagated backwards through the network, starting from the last frame and moving towards the first frame of the video. Note that there is a corner case for the first frame while predicting $y_0$. To circumvent it, we use just duplicate the first frame as the initial prediction to kick-start the training.

## 4 EXPERIMENTS AND RESULTS

### 4.1 DATASETS

As we focus on the problem of generic video enhancement of live videos, presence of multiple degradations is very common. Also, a major use case for real-time video enhancement is video conferencing where the video actually contains the torso/face of the person. To reflect these real-world scenarios, we curate two datasets for the task of video enhancement: i) PortraitVideo and ii) FullVideo.

**PortraitVideo:** We build PortraitVideo on top of TalkingHeads Wang et al. (2021) which is a public dataset. Here, the frame is fixed allowing only the movement of the head to be captured, which

simulates a scenario where the camera is fixed during video calls. The face region is then is cropped similar to face image datasets like FFHQ Karras et al. (2019). Also, we note that TalkingHeads consists of a lot of non-human faces like cartoons and avatars as well. Further, a lot of videos are of very low quality and hence unsuitable for training or evaluation of video restoration. So, we curate PortraitVideo by skipping low quality videos and pick 113 face videos for training and 20 face videos for testing. We fix the resolution of the faces to $384 \times 384$. The videos are processed at 30 frames per second (FPS) with a total of 150 frames per video. We use a mixture of degradations like blur with varying kernels, compression artifacts, noise, small distortions in brightness, contrast, hue, and saturation. The exact details of these degradations can be found in the supplement.

**FullVideo:** We develop this dataset using high quality videos. The video IDs are taken from Talking-Heads dataset however we do not use any of the pre-processing techniques from the TalkingHeads dataset so that the original information of the scene is maintained. We also manually filter to keep only high quality videos. There are 132 training videos and 20 testing videos, and all videos are $720 \times 1280$, 30 FPS and 128 frames long. We apply similar degradations as PortraitVideo for this dataset. The major difference is that this dataset is of a higher resolution and captures more context around the face, including the speaker's body and the rest of the scene.

## 4.2 Implementation Details

We prototype our method using PyTorch on NVIDIA A100 GPU cluster. ReBotNet is trained with a learning rate of $4e^{-4}$ using Adam optimizer, and a cosine annealing learning rate scheduler with a minimum learning rate of $1e^{-7}$. The training is parallelized across 8 NVIDIA A100 GPUs, with each GPU processing a single video. The model is trained for 500,000 iterations. For fair comparison with existing methods, we only use the commonly used Charbonnier loss Barron (2019) to train all models. More configuration details of the architecture can be found in the supplementary.

## 4.3 Comparison with Previous Works

We compare ReBotNet against multiple recent methods. Recurrent Video Restoration Transformer (RVRT) Liang et al. (2022b) is the current SOTA method across many tasks like deblurring, denoising, super-resolution, and video-frame interpolation. We also compare against Video Restoration Transformer (VRT) Liang et al. (2022a), the SOTA convolution-based video super-resolution method BasicVSR++ Chan et al. (2022b), and the fastest deblurring method FastDVD for fair comparison. We retrain all these methods on the new datasets PortraitVideo and FullVideo using their publicly available code.

Initially, we conducted experiments on the new datasets PortraitVideo and FullVideo using the default configurations of VRT, RVRT, BasicVSR++, and FastDVD, as provided in their publicly available code as seen in the first few rows of Table 1. It is important to mention that these models have different levels of floating-point operations (FLOPs). Therefore, to ensure a fair comparison, we assessed the performance of ReBotNet in different FLOP regimes in comparison to the previous methods. This approach helped us gain a comprehensive understanding of the performance of these models across different levels of FLOPs. We pick the embedding dimension across different levels of the network as the hyper-parameter to change the FLOPs Kondratyuk et al. (2021). We acquire different configurations of FLOPs regimes of Small (10Gs), Medium (50Gs), and Large (400Gs). The exact configuration details can be found in the supplementary material. Note that RVRT does not have a S configuration as it is infeasible to scale down the model near 10 GFLOPs due to its inherent design. It should also be noted that for each configuration, we ensured that the computational complexity of ReBotNet remained lower than that of the other models being compared. To provide an example, when evaluating models in the medium regime, we compared ReBotNet, which had a complexity of 56.06, with VRT, which had a complexity of 60.18, and RVRT, which had a complexity of 62.42. In all of our experiments, we used a consistent number of frames, which was set to 2 for all models except for FastDVD, which was designed to process 5 frames. To evaluate the models, we compute the PSNR and SSIM for each individual frame of a video, and then average these values across all frames within the video. We then calculated the mean PSNR and SSIM across all videos and present these results in Table 1. Additionally, we measure the inference time for each method by forwarding 2 frames of dimensions $(384, 384)$ through the network. To obtain

Table 1: Comparison of quantitative results of ReBotNet with previous methods. † represents the default configuration from paper and public code. S, M, L represent the small (∼10G), medium (∼50G), and large (∼400G) FLOPs regimes.

| Method | GFLOPs (↓) | Latency (in ms) (↓) | PortraitVideo | | FullVideo | |
|---|---|---|---|---|---|---|
| | | | PSNR (↑) | SSIM (↑) | PSNR (↑) | SSIM (↑) |
| FastDVDNet † | 367.81 | 36.23 | 28.88 | 0.8516 | 29.56 | 0.8577 |
| VRT † | 2054.32 | 781.15 | 31.70 | 0.8835 | 33.49 | 0.9140 |
| BasicVSR++ † | 157.53 | 49.55 | 31.26 | 0.8739 | 33.10 | 0.9078 |
| RVRT † | 396.29 | 52.30 | 31.92 | 0.8870 | 33.79 | 0.9191 |
| FastDVDNet (S) | 15.85 | 30.51 | 27.97 | 0.8384 | 28.16 | 0.8459 |
| VRT (S) | 15.22 | 48.73 | 30.80 | 0.8681 | 31.85 | 0.8901 |
| BasicVSR++ (S) | 19.05 | 29.08 | 30.90 | 0.8705 | 32.78 | 0.8950 |
| RVRT (S) | - | - | - | - | - | - |
| RebotNet (S) | **13.02** | **13.15** | **31.25** | **0.8778** | **33.45** | **0.9113** |
| FastDVDNet (M) | 64.51 | 33.89 | 28.52 | 0.8405 | 29.35 | 0.8528 |
| VRT (M) | 60.18 | 58.89 | 30.98 | 0.8701 | 32.35 | 0.8987 |
| BasicVSR++ (M) | 60.93 | 41.18 | 31.19 | 0.8729 | 33.04 | 0.9051 |
| RVRT (M) | 62.42 | 35.93 | 31.60 | 0.8821 | **33.59** | 0.9145 |
| RebotNet (M) | **56.06** | **15.02** | **31.85** | **0.8865** | 33.45 | **0.9168** |
| FastDVDNet (L) | 416.90 | 37.14 | 28.93 | 0.8537 | 29.68 | 0.8593 |
| VRT (L) | 419.32 | 91.74 | 31.09 | 0.8729 | 32.68 | 0.9014 |
| BasicVSR++ (L) | 403.22 | 73.32 | 31.40 | 0.8775 | 33.31 | 0.9126 |
| RVRT (L) | 396.29 | 52.30 | 31.92 | 0.8870 | **33.79** | 0.9191 |
| RebotNet (L) | **363.76** | **19.98** | **32.13** | **0.8902** | 33.65 | **0.9199** |

the latency, we perform GPU warm-up for 10 iterations and then feed-forward the clip 1000 times, reporting the average. The latency was recorded on a NVIDIA A100 GPU.

Table 1 demonstrates that our method outperforms most previous approaches in terms of PSNR and SSIM, while using less computational resources across most regimes for both datasets. A significant advantage of our model is its fast inference speed, which is 2.5x faster than the previous best performing method, RVRT. These gains can also been seen in the chart illustrated in Figure 4(a). We also note that we get better results than the original implementations which have way more computations (as seen in first few rows of Table 1). The efficiency of ReBotNet comes because of its effective design while also employing token mixing mechanisms by using mixers. The main contribution towards computation in transformer-based methods like RVRT and VRT come from the self-attention mechanism acting at original scale of the image. Note that we do not use self-attention but replace it with a careful design choice that matches (or even exceeds) its performance.

In Figures 2 and 3, we present qualitative results from PortraitVideo and FullVideo dataset. It can be observed that our method is better than previous methods in terms of quality. The enhanced details are much visible in ReBotnet when compared to other methods. The results are taken from the medium configurations of each model. More results can be found in the appendix.

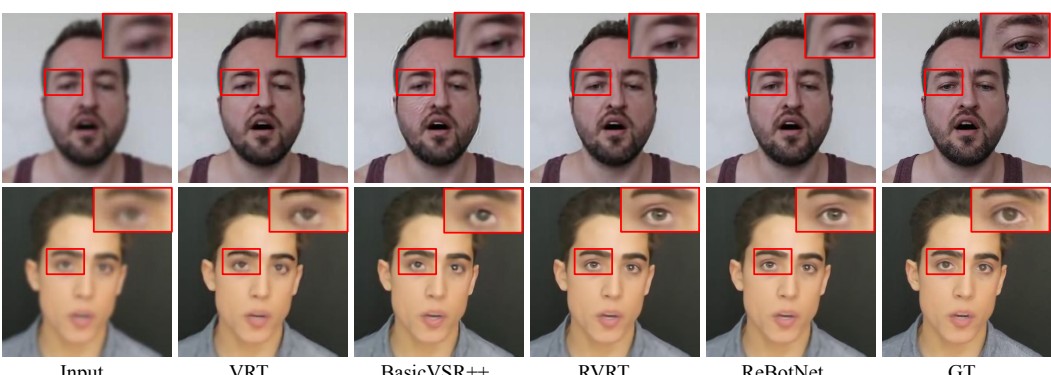

| Input | VRT | BasicVSR++ | RVRT | ReBotNet | GT |

Figure 2: Qualitative Results on *PortraitVideo* dataset. Please zoom in for better visualization.

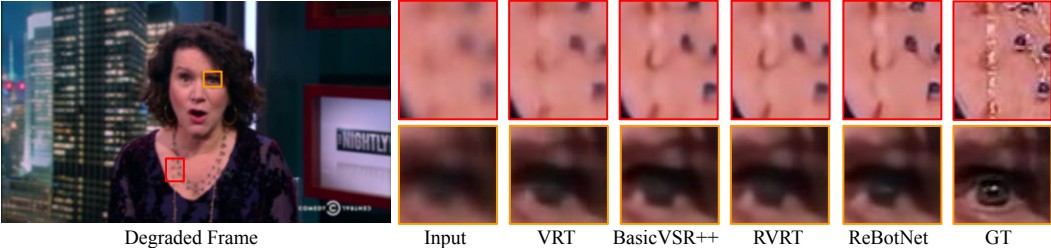

| Degraded Frame | Input | VRT | BasicVSR++ | RVRT | ReBotNet | GT |

Figure 3: Qualitative Results on *FullVideo* dataset. Please zoom in for better visualization.

## 4.4 USER STUDY

To validate the perceptual superiority of ReBotNet for video enhancement, we conducted a user study on the M configuration models on PortraitVideo dataset. We compare our approach to each competing method in a one-to-one comparison. We recruited experts with technical experience in conference video streaming and image enhancement. Each expert evaluated on average 80 video comparisons across four baseline methods. For each comparison, we showed output videos of our method and one competing method, played both videos simultaneously and asked the user to rate which video had a higher quality with the corresponding scores ("much worse", -2), ("worse", 1), ("same", 0), ("better", 1) and ("much better", 2). We calculated the mean score and 95% confidence intervals for paired samples and report them in Table 2. The user study demonstrates the superiority of our method. Despite RVRT being the closest second, our method is still preferred over it while also being more efficient and faster.

Table 2: User study results on PortraitVideo dataset.

| Method | Preference for ReBotNet | 95% Confidence Interval |
|---|---|---|
| FastDVDNet | + 1.83 | 0.059 |
| VRT | + 1.61 | 0.088 |
| BasicVSR++ | + 1.63 | 0.105 |
| RVRT | + 0.08 | 0.073 |

## 5 DISCUSSION

**FPS and Peak Memory Usage:** In Figure 4, we provide a comparison of ReBotNet's frames per second (FPS) rate and peak memory usage with previous methods. For this analysis, we consider feed forward of 2 frames of resolution $384 \times 384$ and consider RebotNet (L) configuration with original implementations for the previous methods. It can be observed that our method has a FPS that is real-time while also not occupying much memory. We note that 30 FPS is considered real-time for applications like video conferencing. Also, ReBotNet has one of the least memory requirements compared to other methods due to its efficient design and implementation.

**Analysis on ReBotNet:** In order to elucidate our design decisions for ReBotNet, we carry out a set of experiments using various parameter configurations, which affect both the performance and computational aspects of the model. These experiments are conducted on the PortraitVideo dataset, using ReBotNet (M) as the base configuration and are reported in Table 3. We analyze the performance along with computation and latency on different configurations of embedding dimension in Mixer (Table 3.a), depth of the bottleneck (Table 3.b), and the number of frames (Table 3.c).

**Ablation Study:** In order to investigate the contribution of each component proposed in the work, we conduct an ablation study using the PortraitVideo dataset. The results of these experiments are shown in Table 4. First, we use the Tubelet tokens extracted from spatio-temporal branch where we use ConvNext encoder directly with a decoder to get the prediction. Then, we consider a configuration where we use image tokens extracted using linear layers from the spatial branch directly forwarded to decoder to get the prediction. This configuration obtains the best latency however suffers from a significant drop in performance. Next, we fuse features extracted from both these branches and use the common decoder. This shows a relative improvement in terms of performance

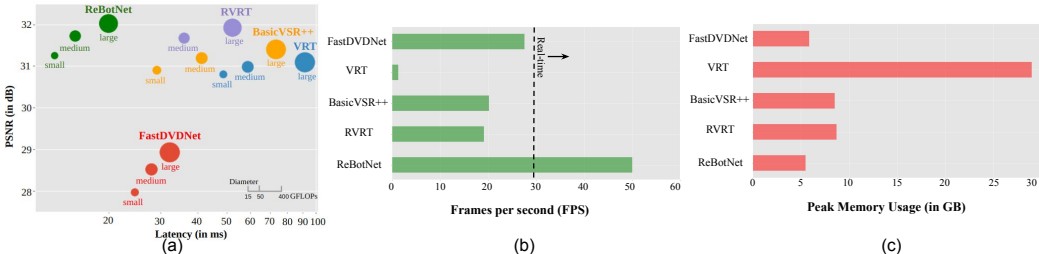

Figure 4: (a) A comparison between the performance of ReBotNet with state-of-the art video restoration networks across different FLOPs regimes on a NVIDIA A100 GPU for PortraitVideo dataset. ReBotNet is observed to give the best performance while having the least latency. (b) Comparison chart of ReBotNet (L) against default configurations of previous methods for Frames Per Second (FPS) and (c) Peak Memory Usage (in GB), as measured on NVIDIA A100 GPU for $2 \times 3 \times 384 \times 384$ input resolution.

Table 3: Analysis on the (a) number of embedding dimension in Mixer (b) depth of the bottleneck (c) number of frames taken.

| Embedding | PSNR (↑) | SSIM (↑) | GFLOPs (↓) | Latency (↓) |
|---|---|---|---|---|
| 128 | 31.79 | 0.8851 | 55.50 | 14.85 |
| 256 | 31.85 | 0.8865 | 56.06 | 15.02 |
| 512 | 31.90 | 0.8869 | 56.60 | 15.27 |
| 728 | 31.89 | 0.8869 | 57.14 | 15.36 |

(a)

| Depth | PSNR (↑) | SSIM (↑) | GFLOPs (↓) | Latency (↓) |
|---|---|---|---|---|
| 2 | 31.83 | 0.8864 | 55.50 | 14.67 |
| 4 | 31.85 | 0.8865 | 56.06 | 15.02 |
| 6 | 31.87 | 0.8866 | 57.14 | 15.31 |
| 8 | 31.81 | 0.8861 | 58.08 | 16.34 |

(b)

| Frames | PSNR (↑) | SSIM (↑) | GFLOPs (↓) | Latency (↓) |
|---|---|---|---|---|
| 1 | 29.56 | 0.8586 | 55.50 | 14.85 |
| 2 | 31.85 | 0.8865 | 56.06 | 15.02 |
| 3 | 31.88 | 0.8871 | 57.14 | 15.16 |
| 4 | 31.92 | 0.8874 | 58.08 | 15.40 |

(c)

without much addition in computation. Note that here the FLOPs of fused configuration is not direct addition between FLOPs of tubulet tokens and image tokens as the decoder's computation was common in both the previous setups. Next, we add the bottleneck mixers which obtains an improvement in performance with little increase in compute. Finally, we add the recurrent training setup which adds no increase in compute but improves the performance. Our findings indicate that each individual component in ReBotNet plays a vital role.

Table 4: Ablation study on PortraitVideo dataset.

| Tubelet Tokens | Image Tokens | Bottleneck Mixer | Recurrent Setup | PSNR (↑) | SSIM (↑) | GFLOPs (↓) | Latency (↓) |
|---|---|---|---|---|---|---|---|
| ✓ | ✗ | ✗ | ✗ | 31.24 | 0.8768 | 54.94 | 14.27 |
| ✗ | ✓ | ✗ | ✗ | 28.01 | 0.8295 | 41.94 | 5.63 |
| ✓ | ✓ | ✗ | ✗ | 31.41 | 0.8792 | 55.50 | 14.67 |
| ✓ | ✓ | ✓ | ✗ | 31.59 | 0.8822 | 56.06 | 15.02 |
| ✓ | ✓ | ✓ | ✓ | 31.85 | 0.8865 | 56.06 | 15.02 |

## 6 CONCLUSION

In this paper, we proposed a novel approach for real-time video enhancement by proposing a new framework: Recurrent bottleneck mixer network (ReBotNet). ReBotNet combines the advantages of both recurrent setup and bottleneck models, allowing it to effectively capture temporal dependencies in the video while reducing the computational complexity and memory requirements. In specific, we showed that an efficient way of encoding temporal and spatial information for videos is by using a combination of convnext encoders to encode low-level information and a bottleneck mixer to encode high-level details. We evaluated the performance of ReBotNet on multiple video enhancement datasets. The results showed that our proposed method outperformed state-of-the-art methods in terms computational efficiency while matching or outperforming them in terms of visual quality. We also conducted an user study with technical experts to validate the usefulness of our proposed network and a clear ablation study to illustrate the contribution of each design choice in our proposed network. Further research could explore the potential applications of the proposed network for other video processing tasks.

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

APPENDIX

## A  CONFIGURATIONS OF REBOTNET

In the main paper, we mentioned we conducted experiments with different FLOPs regimes for all the methods. We did that by controlling the depth of the bottleneck and the embedding dimension of different methods to get the required FLOPs. In Tables 5, 6, and 7 we provide the exact configurations of ReBotNet - S,M, and L respectively. More analysis on the dependence of these parameters were provided in the main paper.

Table 5: Configuration of ReBotNet-S.

| Block | Type | Value |
|---|---|---|
| Branch I | Number of Layers | 4 |
| | Depths per layer | [4,4,4,4] |
| | Embedding dimensions | [28,36,48,64] |
| Branch II | Patch size | 1 |
| | Embedding Dimension | 256 |
| Bottleneck | Depth | 4 |
| | Input Dimension | 64 |
| | Hidden Dimension | 728 |

Table 6: Configuration of ReBotNet-M.

| Block | Type | Value |
|---|---|---|
| Branch I | Number of Layers | 4 |
| | Depths per layer | [4,4,4,4] |
| | Embedding dimensions | [64,80,108,116] |
| Branch II | Patch size | 1 |
| | Embedding Dimension | 256 |
| Bottleneck | Depth | 4 |
| | Input Dimension | 116 |
| | Hidden Dimension | 728 |

Table 7: Configuration of ReBotNet-L.

| Block | Type | Value |
|---|---|---|
| Branch I | Number of Layers | 4 |
| | Depths per layer | [5,5,5,4] |
| | Embedding dimensions | [172,180,188,196] |
| Branch II | Patch size | 1 |
| | Embedding Dimension | 256 |
| Bottleneck | Depth | 4 |
| | Input Dimension | 64 |
| | Hidden Dimension | 728 |

## B  CONFIGURATION OF BASELINES

We used the publicly available codes for the original implementations of FastDVDNet, BasicVSR++, VRT, and RVRT; the results of which can be seen in Table 1 of the main paper. For the S,M and L configurations we use the same configurations of the original implementations but change the embedding dimensions. These changes have been illustrated in Tables 8, 9, and 10. OG means the original implementation. Note that RVRT does not have a S configuration as even with embedding dimensions of $[1, 1, 1]$, the FLOPs does not hit the range of 10 GFLOPs.

Table 8: Configurations of VRT.

| Method | Embedding Dimension |
|--------|---------------------|
| VRT - S | [24,24,24,24,24,24,24,24,24,24] |
| VRT - M | [48,48,48,48,48,48,48,48,48,48] |
| VRT - L | [180,180,180,180,180,180,120,120,120,120] |
| VRT - OG | [180,180,180,180,180,180,120,120,120,120,120,120] |

Table 9: Configurations of RVRT.

| Method | Embedding Dimension |
|--------|---------------------|
| RVRT - S | - |
| RVRT - M | [36,36,36] |
| RVRT - L | [192,192,192] |
| RVRT - OG | [192,192,192] |

Table 10: Configurations of FastDVDNet.

| Method | Embedding Dimension |
|--------|---------------------|
| FastDVDNet - S | [32, 48, 72, 96] |
| FastDVDNet - M | [64, 80, 108, 116] |
| FastDVDNet - L | [96, 112, 132, 144] |
| FastDVDNet - OG | [80, 96, 132, 144] |

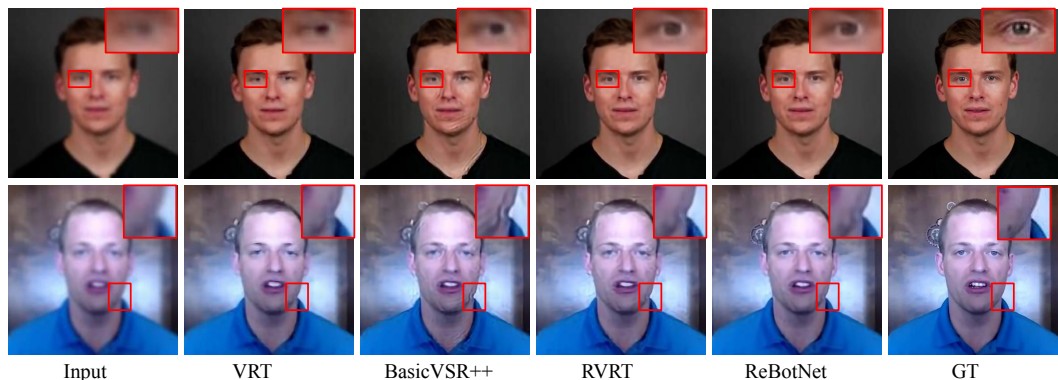

| Input | VRT | BasicVSR++ | RVRT | ReBotNet | GT |
|-------|-----|-----------|------|----------|-----|

Figure 5: Qualitative Results on *PortraitVideo* dataset. Please zoom in for better visualization.

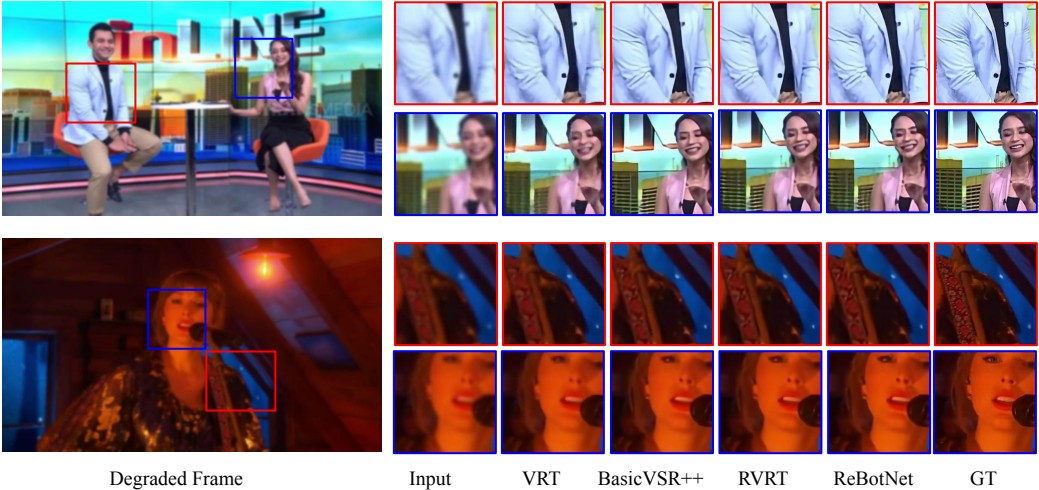

Figure 6: Qualitative Results on *FullVideo* dataset. Please zoom in for better visualization.

## C    EXPERIMENTS ON PURE MIXERS

We observed that MLP-Mixers tend to exhibit a noticeable decline in quality when applied directly for video enhancement compared to transformer-based approaches. Using Mixers directly on large size images still takes a lot of compute and makes it difficult to achieve real-time speed. In Table 11, we conduct an experiment where we take VRT as the base network and convert all the transformer blocks in it to MLP-Mixers. The experiment is conducted on the DVD dataset. It can be observed that the although the computation reduces, the performance also drops significantly. And still the computation is far away from obtaining a real-time FPS. This motivates us to work towards our design of ReBotNet as seen in the main paper.

Table 11: Experiment on pure mixers.

| Method | PSNR | SSIM | GFLOPs | FPS |
|---|---|---|---|---|
| VRT | 34.24 | 0.9651 | 2054.32 | 1 |
| VRT (Mixers) | 32.14 | 0.9429 | 1495.06 | 2 |

## D    MORE QUALITATIVE RESULTS

In Figures 5 and 6, we provide more qualitative results on PortraitVideo and FullVideo datasets respectively.

## E    DEGRADATIONS

In Table 12, we provide the detailed configurations of degradations that we use in PortraitVideo and FullVideo dataset. In all the rows where there is a range, we choose a random value in the range. To get the final degradation of a sample image at hand, we choose a random combination of the degradations from Table 12. These values were decided to emulate degradations possible in real-world and after consulting experts working in the field of video conferencing.

## F    REASONS BEHIND DESIGN CHOICES IN BRANCH I

We pick ConvNext blocks over basic ConvNet blocks as it has been shown that they are both efficient and effective than ConvNets. Each ConvNext block first consists of a depth-wise convolution block

Table 12: Degradations used in PortraitVideo and FullVideo datasets.

| Type of Degradation | Value |
|---|---|
| Eye Enlarge ratio | 1.4 |
| Blur kernel size | 15 |
| Kernel Isotropic Probability | 0.5 |
| Blur Sigma | [0.1,3] |
| Downsampling range | [0.8,2.5] |
| Noise amplitude | [0,0.1] |
| Compression Quality | [70,100] |
| Brightness | [0.8,1.1] |
| Contrast | [0.8,1.1] |
| Saturation | [0.8,1.1] |
| Hue | [-0.05,0.05] |

with kernel size of $7 \times 7$, stride 1 and padding 3. Using a large kernel size is to have a larger receptive field similar to non-local attention. It was observed in Liu et al. (2022b) that the benefit of larger kernel sizes reaches a saturation point at at $7 \times 7$. It is followed by a layer normalization and a point-wise convolution function. The point-wise convolution is basically a convolution layer with kernel size $1 \times 1$. The output of this is activated using GeLU activation and then forwarded to another point-wise convolution to get the output. More details of this why this exact setup is followed can be found in Liu et al. (2022b). We also have downsampling blocks after each level in the ConvNext encoder. The number of ConvNext blocks is a hyperparameter. However, for simplicity we fixed the number of total levels as 4 which means the downsampling is done only 4 times throughout the encoder.

## G  COMPARISON ON PUBLIC DATASETS

We also conduct experiments on single degradation public datasets like DVD, GoPro and report the results in Table 13. For this, we use ReBotNet (L) and compare against the default configurations of previous methods. It can be observed that we obtain a competitive performance in spite of low latency of our model, which can be already seen in Table 1.

Table 13: Comparison of ReBotNet with previous methods on public datasets. Numbers correspond to PSNR / SSIM.

| Method | DVD Su et al. (2017) | GoPro Nah et al. (2017) |
|---|---|---|
| DeepDeblur Nah et al. (2017) | 29.85 / 0.8800 | 38.23 / 0.9162 |
| EDVR Wang et al. (2019) | 31.82 / 0.9160 | 31.54 / 0.9260 |
| TSP Pan et al. (2020) | 32.13 / 0.9268 | 31.67 / 0.9279 |
| PVDNet Son et al. (2021) | 32.31 / 0.9260 | 31.98 / 0.9280 |
| VRT Liang et al. (2022a) | 34.24 / 0.9651 | 34.81 / 0.9724 |
| RVRT Liang et al. (2022b) | 34.30 / 0.9655 | 34.92 / 0.9738 |
| ReBotNet | 34.30/ 0.9656 | 34.90 / 0.9734 |

## H  EXPERIMENTS LEADING TO OUR DESIGN CHOICE

We experimented with ConvNets, ConvNexts, Transformers, and Mixers individually before finalizing our design choice—ReBotNet. These experiments yielded analytical insights crucial in shaping our decision." FastDVDNet and BasicVSR++ represent purely convolutional methods, whereas VRT and RVRT utilize transformer-based approaches with transposed convolution layers in their decoders. Despite their unique design improvements aimed at enhancing performance, these methods share a common encoder-decoder architecture, typical of most restoration techniques. To comprehend the advantages of encoder design, we conduct an experimental setup altering the encoder design while maintaining a fixed decoder with simple transposed convolution layers. Our objective is to match the performance of these models, aiming to gauge the computational requirements needed to achieve equivalent performance among these diverse encoders. The performance in terms

of PSNR on the Portrait Video dataset and the latency of each of these models in terms of ms is reported in the table 14 below:

Table 14: Experiments with different encoders on PortraitVideo dataset.

| Encoder | Latency (in ms) | PSNR |
|---|---|---|
| ConvNet | 86.89 | 31.24 |
| ConvNext | 30.56 | 31.22 |
| Transformer | 140.58 | 31.25 |
| Mixer | 28.65 | 31.23 |

We can observe that while Transformers take more inference time and complexity to match ConvNet's performance while both ConvNext and Mixers are more efficient than their predecessors.

After considering these insights, our focus shifted towards leveraging the strengths of both ConvNext and Mixer. Combining a convolutional encoder for initial feature extraction with a mixer network in the bottleneck has demonstrated effectiveness. Convolutional layers excel in efficient low-level feature extraction, while mixers, following transformers, showcase strong representative abilities in extracting deep features, as also highlighted in Srinivas et al. (2021). We performed experiments where we slowly introduce Mixer in the bottleneck of ConvNext which can be seen in the following table 15 where the PSNR is reported on PortraitVideo dataset:

Table 15: Experiments on PortraitVideo dataset with difference configurations of ConvNext and Mixer at different encoder blocks.

| ConvNext | Mixer | Latency (in ms) | PSNR |
|---|---|---|---|
| 1,2,3,4,5 | - | 30.56 | 31.24 |
| 1,2,3 | 4,5 | **15.58** | 31.25 |
| 1,2 | 3,4,5 | 18.79 | 31.25 |
| 1 | 2,3,4,5 | 26.77 | 31.23 |
| - | 1,2,3,4,5 | 28.65 | 31.23 |
| 4,5 | 1,2,3 | 36.85 | 31.25 |

This illustrates our empirical journey towards the core design element of ReBotNet—a convolutional feature extractor followed by a mixer bottleneck. Notably, the last row presents an alternative configuration with early mixer layers and bottleneck ConvNext layers, resulting in sub-optimal performance. These analytical experiments were pivotal in finalizing the design choice of ReBotNet.

## I    DIFFERENCE FROM EXISTING DATASETS

We tabulate the difference of our curated datasets with existing datasets in the following Table 16.

Table 16: Comparison of newly curated datasets with previous dataests.

| | PortraitVideo/FullVideo | Existing Datasets |
|---|---|---|
| High Quality | Y | Y |
| Live Scenarios | Y | N |
| Focus on Humans | Y | N |
| Multiple Degradations | Y | N |

## J    TEMPORAL CONSISTENCY

We would like to note that quantitatively evaluating temporal consistency is an actively researched field. With that being said, we report the difference in SSIM between consecutive frames to quantitatively evaluate the temporal consistency. The following table 17 is the mean SSIM difference across consecutive frames for all videos in the Portrait Video dataset. Differences in SSIM can give us details about how smooth the transitions are temporally where low difference means temporally

consistent cases. It can be seen that the SSIM difference is less for the configuration including temporal branch proving the usefulness of our method.

Table 17: SSIM difference reported to understand the temporal consistency aspect of the introduced temporal branch.

| Config | SSIM Difference |
|---|---|
| Ground Truth | 0.104 |
| Without Temp. Branch | 0.158 |
| With Temp. Branch | 0.124 |

