# OpenReview forum: "ReBotNet: Fast Real-time Video Enhancement"
_ICLR.cc/2024/Conference — Submitted to ICLR 2024_

### Official Review · Reviewer_ki94 · 2023-10-23

**Soundness:** 3 good
**Presentation:** 3 good
**Contribution:** 2 fair
**Rating:** 6
**Confidence:** 4

**Summary:**

The paper introduces a fast and efficient framework titled Recurrent Bottleneck Mixer Network (ReBotNet) for performing real-time video enhancement. This can carry practical applications in areas such as live video calls and video streams. The novelty of ReBotNet lies in its dual-branch system. The first branch utilizes a ConvNext-based encoder to learn spatio-temporal features by tokenizing the input frames along the spatial and temporal dimensions. These tokens are then processed with a bottleneck mixer. The second branch enhances temporal consistency by directly employing a mixer on tokens extracted from individual frames. The branches converge, with a common decoder merging the features to predict the enhanced frame.

Additionally, the authors use a recurrent training approach where the prediction of the last frame is utilized to efficiently improve the current frame while enhancing temporal consistency. The effectiveness of this method is evaluated on two newly curated datasets representing real-world video calls and streaming scenarios. The results obtained indicate that ReBotNet outperforms existing techniques with less computation, minimal memory requirements, and faster inference time.

**Strengths:**

1. Overall, the method proposed in this paper is novel and unique. Previous methods usually embed optical flow estimation explicitly or implicitly.
2. The goal of the paper is to propose a real-time video enhancement model, which I think has practical value.
3. The comparative experiments in the paper include different computational complexity levels and user studies.
4. The authors provide two new datasets.

**Weaknesses:**

1. The paper only conducts experiments on the newly proposed datasets. But these two datasets are not larger or more extensive than previous datasets, so I worry that the experiments will not be convincing enough.
2. The paper does not seem to have submitted a demonstration video. Judging from the pictures in the paper, the improvement in visual effects is not significant.

**Questions:**

1. "Unlike these works that require compute intensive optical flow, we develop a simple and efficient frame-recurrent setup with low computational overhead." Optical flow calculations do not seem to be necessarily linked to high computational overhead. Refer to "Optical flow estimation using a spatial pyramid network" or "Real-time intermediate flow estimation for video frame interpolation".
2. "A major use case for real-time video enhancement is videoconferencing where the video actually contains the torso/face of the person. " This sentence does not seem to be enough to support the paper's experimentation in this scenario only. I am particularly worried that the background of these scenes is static, and the movement of faces is different from ordinary objects, making it impossible to judge the generalization of the proposed method in general scenes.
Even though real-time video conferencing is an important requirement, low-overhead video enhancement makes sense for many other users.
3. The author mentioned "The training is parallelized across 8 NVIDIA A100 GPUs, with each GPU processing a single video. ", is this training method fair to other models?

---

> ### Author Response · Authors · 2023-11-21
> **Official Response to Reviewer ki94**
>
> We thank the reviewer for their comments. We are encouraged by their positive comments on novelty, uniqueness, and validating our practical compute-efficient solution. In what follows, we address all the concerns:
>
> ## 1) Comparison with other datasets:
>
> We have now added results on two more public datasets where our performance is similar to that of RVRT while being much faster and obtaining real-time settings. It can be seen in the table below:
>
> |   Method   |  DVD  | GoPro |
> |:----------:|:-----:|:-----:|
> | DeepDeblur | 29.85 | 38.23 |
> |   DVDNet   | 32.31 | 31.98 |
> |     VRT    | 34.24 | 34.81 |
> |    RVRT    | 34.30 | 34.92 |
> |  ReBotNet  | 34.30 | 34.90 |
>
> We would also like to point out that our curated datasets are comparable in size to the existing datasets while dealing with more challenging cases due to multiple degradations operating on the same video. Also, it emulates live video call and live streaming scenarios unlike existing datasets.
>
> ## 2) Demonstration Videos:
>
> We provide some demonstration videos of comparison with previous methods as well as direct slider comparison with real videos in [this link](https://anonymous.4open.science/r/iclr_rebuttal2-EB61/Comparison1.mov). Please look at the left hand side to access all the files from the window that pops up from this link. We are worried right now that releasing a lot of demonstration videos might have a problem with protecting our anonymity. We promise to release a link to more demonstration videos after the reviews.
>
> ## 3) Optical Flow and Computation:
>
> We apologize to the reviewer for that sentence and thank the reviewer for pointing out those papers. We have removed them in our revised version but highlight that our contributions or work does not have any effect from this as our method takes a completely different approach and provides a working solution for the problem of video enhancement.
>
> ## 4) Enhancement in Dynamic Scenes:
>
> While we understand the reviewer’s concerns on static scenes, most of the videos in our FullVideo dataset PortraitVideo in fact have dynamic scenes. This is reflected in the demo videos we have now linked (in point 2). In the paper, while we mention  “video actually contains the torso/face of the person”, we do not say that the background is always static. The background can be dynamic and we have included those cases in our datasets as well.
>
> ## 5) Fair Comparison with existing methods:
>
> During our experimental setup, we made sure to look into the code-bases of previous methods like RVRT, VRT, and BasicVSR++ and found that they also train the network across 8 GPUs for their experiments making our comparison with those methods fair.

---

> ### Comment · Reviewer_ki94 · 2023-11-22
>
> After reading the rebuttal and other reviews, I generally support the acceptance of this paper and raised my rating. I hope the author can continue to push the results of the paper into applications.
>
> Because of the real-time goal, similar papers often cannot adopt designs that look fancy but have poor hardware support. This limits the author to make the paper "look novel". After reading the paper carefully, I feel that the overall contribution of this paper is worthy of recognition. The extreme efforts towards computational efficiency make this paper different from many previous works.

---

> > ### Author Response · Authors · 2023-11-22
> > **Official Reply to Reviewer ki94**
> >
> > We thank the reviewer for supporting acceptance; understanding and appreciating the uniqueness/novelty of our work.

---

### Official Review · Reviewer_PgtF · 2023-10-29

**Soundness:** 2 fair
**Presentation:** 3 good
**Contribution:** 2 fair
**Rating:** 5
**Confidence:** 4

**Summary:**

This work aims to break through the limitations of existing video enhancement methods in terms of processing speed and introduces a real-time video enhancement method designed for video calls and streaming scenarios. The authors achieve this by constructing a dual-branch framework, each addressing spatial and temporal feature information separately, and then merging them to obtain the final result. A recursive training strategy is further proposed to effectively utilize the predictive information from the previous frame to enhance the prediction quality of the current frame. Additionally, the authors introduce two new video enhancement datasets based on existing data to assist in evaluation.

**Strengths:**

1. The author provides a clear explanation of the motivation behind network architecture and training strategy design.
2. The experimental validation section is relatively comprehensive, and the description of relevant settings is sufficiently detailed.

**Weaknesses:**

1. Although the author has conducted a thorough analysis of the constructed network in terms of details and motivations, it appears challenging to avoid the fact that the techniques used in this work seem to be readily available. It is hoped that the author can further refine the contributions and strengths of the proposed method in this regard. In other words, there may still be room for improvement in terms of innovation in this work.
2. The author uses a direct addition approach to achieve fusion after the two branches, and I'm curious whether the author has tried other methods to better integrate information related to tubelet tokens and image tokens. In other words, I hope the author can provide a brief explanation or analysis of their choice of fusion method.
3. The author's introduction to the curated dataset is not sufficiently clear. It is recommended that the author provide a more intuitive comparison between the curated dataset and existing datasets in terms of data quantity, types of data degradation, and whether they are paired, possibly in the form of a table. Some visual examples should also be included in the manuscript. Additionally, the author does not seem to mention whether these two datasets will be made open-source, which has a certain impact on the contribution of this work.
4. The scenarios targeted by this work are closely related to everyday life. I am quite curious about the performance of the proposed method on some real video data captured using mobile devices. Could the author possibly add relevant experimental results to more comprehensively validate the effectiveness of the proposed method?
5. There appear to be some typographical errors in the manuscript. In the analysis of ReBoNet in Section Five, the author mentions "Table 3 illustrates these results where gray rows correspond to ...", but there doesn't seem to be any gray rows in Table 3. We hope the author can carefully review the manuscript to prevent such situations from occurring.

**Questions:**

Please refer to the Weaknesses.

---

> ### Author Response · Authors · 2023-11-21
> **Official Response to Reviewer PgtF**
>
> We thank the reviewer for the comments and address their concerns below:
>
> ## 1) Contributions:
>
> We highlight our important contributions here:
>
> * proposing the first video enhancement network that can work in real-time settings (more than 30 FPS) and with a good performance.
> * curating two new datasets that resemble actual real-world settings like live video calls and video streams.
> * developing an architecture with ConvNext feature extractors and bottleneck Mixers and showing that this configuration has the best latency-performance tradeoff for video enhancement networks.
> * ensuring temporal consistency by integrating with other techniques like recurrent training and temporal bottleneck branch.
>
> To address the particular point of adding more innovation to this work, we would like to point out that our goal for this work was to obtain a strong video enhancement that works in real-time settings rather than solely introducing novelty. Previous works like RVRT provided more than satisfactory performance for our video enhancement but had an issue with latency which was our main motivation as we targetted live video call enhancement. With ReBotNet, we solve this issue and provide the community with a low-compute model that can also perform well; and our goal was accomplished. With that being said, our work is still novel in its architecture design our architecture is completely different from any of the works in the literature for video enhancement.
>
> ## 2) Integrating Tubelet and Image Tokens:
>
> We did explore alternate techniques for the feature fusion, but found out that a simple addition was the most compute effective option. The performance improvements we obtained from more complicated processes were not significant enough for us to consider them as the fusion strategy. We compare the other alternatives we considered in the following table on PortraitVideo dataset:
>
> |   Method  | Latency |  PSNR |
> |:---------:|:-------:|:-----:|
> |  Addition |  13.15  | 31.25 |
> |   Concat  |  16.85  | 31.25 |
> | Attention |  17.58  | 31.28 |
>
> It can be seen that although using attention to fuse the features did provide some performance improvement, the model was suboptimal in terms of latency, which is our main focus for this work.
>
> ## 3) Difference with previous datasets:
>
> We tabulate the difference of our curated datasets with existing datasets in the following:
>
> |                       | PortraitVideo/FullVideo | Existing Datasets |
> |:---------------------:|:-----------------------:|:-----------------:|
> |      High Quality     |            Y            |         Y         |
> |     Live Scenarios    |            Y            |         N         |
> |    Focus on Humans    |            Y            |         N         |
> | Multiple Degradations |            Y            |         N         |
>
> We will release the dataset by releasing a script to curate and generate the videos from the web similar to TalkingHead dataset.
>
> ## 4) Everyday life scenarios:
>
> To make the model work on everyday things like mobile phone videos, we need to scale the data that we train the model on. We consider this our future work. Our aim would be to ensure that the model's training data comprehensively encompasses diverse and representative samples from everyday scenarios, enabling its effective application to diverse content. Currently, our contribution is more towards the network architecture and not towards an generalizable model. In fact, this is also an active field of research in video restoration.
>
> ## 5) Table 3:
>
> We thank the reviewer for pointing out this error and we have corrected it in the revised version.

---

### Official Review · Reviewer_e4UV · 2023-10-30

**Soundness:** 3 good
**Presentation:** 3 good
**Contribution:** 1 poor
**Rating:** 5
**Confidence:** 5

**Summary:**

This paper presents ReBotNet, designed for real-time video enhancement. The proposed dual-branch system utilizes spatio-temporal tokenization of frames and combines features from both branches to improve the output. A recurrent framework is employed to include previous frame predictions, ensuring better temporal consistency. The methods have been tested on newly curated datasets, demonstrating state-of-the-art results.

**Strengths:**

The effectiveness of the proposed method was demonstrated through testing on two specially created datasets, mimicking real-world video scenarios. The results reveal that ReBotNet surpasses existing methods, offering faster performance, less computation, and minimized memory use. ReBotNet can be significantly useful in practical applications. The authors also tried to conduct a fair evaluation by optimizing the results of previous research.

**Weaknesses:**

There are few weaknesses observed in this paper.

1. The novelty of this research compared to other studies is not clear, because there have been many studies on the two-branch framework that processed video cubes with two different temporal dynamics as input.

2. Rather, it seems that the authors have optimized module that were already working well, and it does not appear that a comprehensive experiment has been conducted to justify their approach, by elucidating the importance of the optimized modules shared with the previous studies, if any.

3.  Since the paper focuses on practicality, it is necessary to show experiment results on more open datasets to demonstrate its more general applicability.

4. Although they claimed to have created a video dataset with real-world nose, the experiment only compared using only the video restoration task and a few simple metrics, failing to effectively demonstrate its significance.

**Questions:**

It would be appreciated if the authors could resolve the reviewer's concerns above.

**Details Of Ethics Concerns:**

.

---

> ### Author Response · Authors · 2023-11-21
> **Official Response to Reviewer e4UV**
>
> We thank the reviewer for the comments. We are glad that the reviewer pointed out several useful aspects of our work. In what follows, we clear all the 4 points the reviewer had in the weakness section:
>
> ## 1) Novelty of our work:
>
> We agree with the reviewer that there are other works presenting dual-branch networks but we would like to highlight that we never pointed out our key novelty to be the dual-branch part. Our novelty lies in:
>
> * proposing the first video enhancement network that can work in real time settings with a good performance. Please note that all the previous networks proposed do not focus on this aspect at all and so have computationally complex solutions focussed only on performance making them unsuitable for real-time applications.
>
> *  developing an architecture with ConvNext feature extractors and bottleneck Mixers and showing that this configuration has the best latency-performance tradeoff for video enhancement.
>
> * curating two new datasets that resemble actual real-world settings like live video calls and video streams and also perform user studies to prove the effectiveness of our method.
>
> ## 2) Comprehensive experiments leading to our design choice:
>
> We experimented with ConvNets, ConvNexts, Transformers, and Mixers individually before finalizing our design choice for ReBotNet. These experiments yielded analytical insights crucial in shaping our decision. We started our experiments with a setup altering the encoder design while maintaining a fixed decoder with simple transposed convolution layers. Our objective was to match the performance of these models, aiming to gauge the computational requirements needed to achieve equivalent performance among these different encoders. The performance in terms of PSNR on the Portrait Video dataset and the latency is reported in the table below:
>
> |   Encoder   | Latency (in ms) |  PSNR |
> |:-----------:|:---------------:|:-----:|
> |   ConvNet   |      86.89      | 31.24 |
> |   ConvNext  |    **30.56**    | 31.22 |
> | Transformer |      140.58     | 31.25 |
> |    Mixer    |    **28.65**    | 31.23 |
>
>
> We can observe that while Transformers take more inference time and complexity to match ConvNet’s performance while both ConvNext and Mixers are more efficient than their predecessors.
>
> After considering these insights, our focus shifted towards leveraging the strengths of both ConvNext and Mixer. Combining a convolutional encoder for initial feature extraction with a mixer network in the bottleneck has demonstrated effectiveness. Convolutional layers excel in efficient low-level feature extraction, while mixers, following transformers, showcase strong representative abilities in extracting deep features, as also highlighted in Srinivas et al. (2021). We performed experiments where we slowly introduce Mixer in the bottleneck of ConvNext which can be seen in the following table where the PSNR is reported on PortraitVideo dataset:
>
> |  ConvNext |   Mixer   | Latency (in ms) |  PSNR |
> |:---------:|:---------:|:---------------:|:-----:|
> | 1,2,3,4,5 |     -     |      30.56      | 31.24 |
> |   1,2,3   |    4,5    |    **15.58**    | 31.25 |
> |    1,2    |   3,4,5   |      18.79      | 31.25 |
> |     1     |  2,3,4,5  |      26.77      | 31.23 |
> |     -     | 1,2,3,4,5 |      28.65      | 31.23 |
> |    4,5    |   1,2,3   |      36.85      | 31.25 |
>
> Notably, the last row presents an alternative configuration with early mixer layers and bottleneck ConvNext layers, resulting in sub-optimal performance.  We also point the reviewer to Table 4 of the main paper where we conduct further analysis validating our design choice.
>
> ## 3) Performance on open datasets:
>
> We perform experiments on two public restoration datasets DVD and GoPro and tabulate the results below. It can be observed that ReBotNet performs as good as RVRT while having low latency and so enabling real-time use-cases.
>
> |   Method   |  DVD  | GoPro |
> |:----------:|:-----:|:-----:|
> |     VRT    | 34.24 | 34.81 |
> |    RVRT    | 34.30 | 34.92 |
> |  ReBotNet  | 34.30 | 34.90
>
> ## 4) Failing to effectively demonstrate its significance?
>
> We respectfully disagree with the reviewer's comment on this and emphasize that our datasets simulate challenging real-world scenarios by combining multiple degradations unlike previous works in this literature. These experiments simulate multiple video restoration tasks due to the diversity of the degradations. Also, PSNR and SSIM metrics are the widely used metrics in restoration literature. Mainly, we focus on the latency, which is also not demonstrated in prior literature and demonstrates the significance of our contributions.
>
> Our work also includes a user study revealing where our predictions were found to be more favorable to users, as detailed in Section 4.4 which is an effective way of evaluation. We kindly request the reviewer to provide additional guidance on how to effectively showcase the significance further and we are happy to work on it.

---

> > ### Comment · Reviewer_e4UV · 2023-11-22
> >
> > Thank you for the reviewers' response. The reviewer agrees that real-time processing in video enhancement is an important topic. However, the proposed method seems to be more about combining existing modules and empirically selecting the best combination (the best position/order of convnext and mixer, etc), rather than having a new approach or principle to replace them. Although the optimized results would be technically helpful from a practical perspective, it still seems lacking the novelty in the proposed method.

---

> > > ### Author Response · Authors · 2023-11-22
> > > **Official Reply to Reviewer e4UV**
> > >
> > > We thank the reviewer for the reply and recognizing that our work would be technically helpful from a practical perspective. While we understand that we are not proposing a architecture like transformer or convnet, this work also imparts new knowledge to the community and contributes by:
> > >
> > > * Proposing one of the first network configurations that can obtain a good latency-performance tradeoff for real-time video enhancement.
> > >
> > > * Showing early ConvNext layers are good low-level feature extractors; better than transformers/mixers.
> > >
> > > * Showing bottleneck Mixers are effective at extracting strong deep features; better than ConvNets/ConvNexts.
> > >
> > > * Curating two new datasets emulating real-world scenarios and bench-marking them in real-time settings.
> > >
> > > Please note that our empirical analysis is also new knowledge that we impart to the community and helps understand the usefulness of networks in the realm of video enhancement.

---

### Official Review · Reviewer_SWLx · 2023-10-31

**Soundness:** 3 good
**Presentation:** 3 good
**Contribution:** 3 good
**Rating:** 5
**Confidence:** 4

**Summary:**

This paper presents an efficient video enhancement framework. The network architecture utilizes ConvNext for Spatial-Temporal tokenization of the video and employs a Mixer structure to process the tokens. To evaluate the algorithm, the method introduces two datasets, on which the algorithm performs well (and better than baselines). Overall, it is an effective framework for video enhancement and does make some contributions. However, it is difficult to pinpoint the novelty of the algorithm. Currently, I find it hard for me to make a final decision.

**Strengths:**

+: Well-written. I can clearly understand the design motivations behind most parts.

+: On the two datasets, this algorithm achieves a good balance between efficiency and effectiveness, surpassing many previous algorithms.

+: There is some ablation study to analyze the role of each branch.

**Weaknesses:**

-: The explanation of the contribution to efficiency and effectiveness is not clear enough. The introduction mentions that the combination of ConvNext and Mixer avoids quadratic complexity and guarantees performance. Does it mean that using Mixer speeds up the process (avoiding quadratic complexity), while ConvNext ensures performance? Is the fundamental reason for the acceleration avoiding quadratic complexity?

-: The PSNR and SSIM results are similar to RBRT.

-: The evaluation of temporal consistency seems lacking in the paper. For instance, although the abstract claims that the second branch improves temporal consistency, there are no experiments to support this result.

-: I believe the ablation study is not detailed enough. Combining ConvNext and Mixer is a direct and straightforward idea. In the process of parameter tuning, what analytical experiments are worth providing to the community? I expect to see many detailed experiments on this aspect. Can we use other backbones to replace the ConvNext?

**Questions:**

-: Creating their own dataset is good, but why not compare it with public datasets and other baselines at the same time? Theoretically, this model can also be run and compared on other datasets, right?

---

> ### Author Response · Authors · 2023-11-21
> **Official Response to Reviewer SWLx**
>
> We thank the reviewer for the review and comments. We have addressed all the comments in the following:
>
>
>
> ## 1) Analytical Experiments leading to our design choice:
>
> To comprehend the advantages of encoder design, we started our experiments with a setup altering the encoder design while maintaining a fixed decoder with simple transposed convolution layers. Our objective was to match the performance of these models, aiming to gauge the computational requirements needed to achieve equivalent performance among these different encoders. The performance in terms of PSNR on the Portrait Video dataset and the latency is reported in the table below:
>
> |   Encoder   | Latency (in ms) |  PSNR |
> |:-----------:|:---------------:|:-----:|
> |   ConvNet   |      86.89      | 31.24 |
> |   ConvNext  |    **30.56**    | 31.22 |
> | Transformer |      140.58     | 31.25 |
> |    Mixer    |    **28.65**    | 31.23 |
>
>
> We can observe that while Transformers take more inference time and complexity to match ConvNet’s performance while both ConvNext and Mixers are more efficient than their predecessors.
>
> After considering these insights, our focus shifted towards leveraging the strengths of both ConvNext and Mixer. Combining a convolutional encoder for initial feature extraction with a mixer network in the bottleneck has demonstrated effectiveness. Convolutional layers excel in efficient low-level feature extraction, while mixers, following transformers, showcase strong representative abilities in extracting deep features, as also highlighted in Srinivas et al. (2021). We performed experiments where we slowly introduce Mixer in the bottleneck of ConvNext which can be seen in the following table where the PSNR is reported on PortraitVideo dataset:
>
> |  ConvNext |   Mixer   | Latency (in ms) |  PSNR |
> |:---------:|:---------:|:---------------:|:-----:|
> | 1,2,3,4,5 |     -     |      30.56      | 31.24 |
> |   1,2,3   |    4,5    |    **15.58**    | 31.25 |
> |    1,2    |   3,4,5   |      18.79      | 31.25 |
> |     1     |  2,3,4,5  |      26.77      | 31.23 |
> |     -     | 1,2,3,4,5 |      28.65      | 31.23 |
> |    4,5    |   1,2,3   |      36.85      | 31.25 |
>
> Notably, the last row presents an alternative configuration with early mixer layers and bottleneck ConvNext layers, resulting in sub-optimal performance. These analytical experiments were pivotal in finalizing the design choice of ReBotNet.
>
> ## 2) Key Takeaways / Influence of ConvNext and Mixers:
>
> Expanding on the tables previously discussed, the key takeaways for the community are:
>
> * Early ConvNext layers are good low-level feature extractors; better than transformers/mixers.
> * Bottleneck Mixers are effective at extracting strong deep features; better than ConvNets/ConvNexts.
> * Temporal branch adds temporal consistency and makes inter-frame transitions smoother.
> * A combination of ConvNext feature extractors and bottleneck Mixers proves highly effective for video enhancement, offering an optimal tradeoff between latency and performance.
>
> ## 3) Comparison with RVRT:
>
> Although ReBotNet's performance is on par with RVRT, the primary difference lies in the model's latency. Unlike RVRT, which is unsuitable for real-time applications and remains challenging to scale down while maintaining a favorable latency-to-performance tradeoff (as demonstrated in Table 1), ReBotNet is an entirely different architecture design. This core difference grants ReBotNet the capability for real-time processing, facilitating its effective utilization in live calls and streams, while delivering comparable performance to RVRT.
>
> ## 4) Temporal Consistency:
>
> We would like to note that quantitatively evaluating temporal consistency is an actively researched field. Usually, one is able to evaluate temporal consistency by just looking at the video. We attach a [sample predictions link](https://anonymous.4open.science/r/ICLR_rebuttal-4D33/temporalbranch.mov) with and without the temporal branch and one can see that the one with the temporal branch is smoother with transitions in between frames (we recommend the reviewer to focus on the eye region to clearly see the difference).
>
> We quantify temporal consistency via SSIM differences between consecutive frames in the Portrait Video dataset. The configuration with the temporal branch shows lower differences, indicating improved smoothness.
>
> |        Config        | SSIM Difference |
> |:--------------------:|:---------------:|
> |     Ground Truth     |      0.104      |
> | Without Temp. Branch |      0.158     |
> |   With Temp. Branch  |      0.124     |
>
>
> ## 5) Performance on other datasets:
>
> We perform experiments on two public restoration datasets DVD and GoPro and tabulate the results below. It can be observed that ReBotNet performs as good as RVRT while having low latency and so enabling real-time use-cases.
>
> |   Method   |  DVD  | GoPro |
> |:----------:|:-----:|:-----:|
> |     VRT    | 34.24 | 34.81 |
> |    RVRT    | 34.30 | 34.92 |
> |  ReBotNet  | 34.30 | 34.90 |

---

> > ### Comment · Reviewer_SWLx · 2023-11-22
> >
> > Thank the authors for their detailed explanation. The feedback from the authors addresses most of my concerns.
> >
> > Generally speaking, this paper proposes an efficient approach for video enhancement that achieves on-part performance with SOTA on public datasets and better performance on their proposed dataset. The takeaways for ConvNext and Mixer are good but/and somehow empirical. It is hard to know whether this observation can be applied to other tasks.
> >
> > I believe more discussion with other reviewers should be included to make a fair decision.

---

> > > ### Author Response · Authors · 2023-11-22
> > > **Official Comment to Reviewer SWLx**
> > >
> > > We thank the reviewer for reading and replying to our comments. We are happy that the reviewer found our comments to address most of the concerns.
> > >
> > > We highlight that our on-par or better performance was achieved with very less compute compared to the existing methods. Our primary objective centered on video enhancement, making us to focus our analysis on a single task to achieve greater depth rather than dispersing our efforts.

---

### Meta-Review · Area_Chair_HScQ · 2023-12-05

**Metareview:**

Overall, the work tends to receive a negative overall rating, with three out of four reviewers providing scores slightly below the acceptance threshold. After a thorough examination of the reviewers' comments and the responses provided by the authors, there is a consensus among the three reviewers leaning towards rejection that there might be some room for improvement in the novelty of the proposed method. In other words, from the reviewers' perspective, the method proposed by the authors seems to rely somewhat more on empirical approaches to network construction for specific tasks. While some reviewers acknowledge the contributions and practical value of the work, there may not be sufficiently convincing explanations from the authors regarding the general concerns about novelty. In summary, the work may not have met the standards for ICLR 2023, and I decide to reject it.

**Justification For Why Not Higher Score:**

Please refer to the metareview.

**Justification For Why Not Lower Score:**

N/A

---

### Decision · Program_Chairs · 2024-01-16

Reject